# What are the best systems? New perspectives on NLP Benchmarking

**Pierre Colombo**
L2S
CentraleSupelec, France
pierre.colombo@centralesupelec.fr

**Nathan Noiry**
S2A
Telecom Paris, France
nathan.noiry@telecom-paris.fr

**Ekhine Irurozki**
S2A
Telecom Paris, France
ekhine.irurozki@telecom-paris.fr

**Stephan Clémençon**
S2A
Telecom Paris, France
stephan.clemençon@telecom-paris.fr

## Abstract

In Machine Learning, a benchmark refers to an ensemble of datasets associated with one or multiple metrics together with a way to aggregate different systems performances. They are instrumental in *(i)* assessing the progress of new methods along different axes and *(ii)* selecting the best systems for practical use. This is particularly the case for NLP with the development of large pre-trained models (*e.g.* GPT, BERT) that are expected to generalize well on a variety of tasks. While the community mainly focused on developing new datasets and metrics, there has been little interest in the aggregation procedure, which is often reduced to a simple average over various performance measures. However, this procedure can be problematic when the metrics are on a different scale, which may lead to spurious conclusions. This paper proposes a new procedure to rank systems based on their performance across different tasks. Motivated by the social choice theory, the final system ordering is obtained through aggregating the rankings induced by each task and is theoretically grounded. We conduct extensive numerical experiments (on over 270k scores) to assess the soundness of our approach both on synthetic and real scores (*e.g.* GLUE, EXTREM, SEVAL, TAC, FLICKR). In particular, we show that our method yields different conclusions on state-of-the-art systems than the mean-aggregation procedure while being both more reliable and robust.

## 1 Introduction

This paper is about improving current practices regarding benchmarks of NLP systems. As pointed out by [83], benchmarks are made of datasets, metrics, and a way to aggregate performance. Our point is that, if the bulk of the NLP community efforts on this domain is about collecting new datasets and introducing new metrics, little work is concerned with the third part, namely *how to aggregate various performances*.

**Why are benchmarks vital?** Research advances in Machine Learning (ML) are crucially fueled by reliable evaluation procedures [34, 78]. The latter are indeed mandatory to fairly compare new methods and systems. Usually, one relies on a well-chosen metric that reflects the ability to perform on a task – *e.g.* accuracy for classification, mean-squared error for regression.

**The multi-tasks evaluation setting.** If single-tasks problems are quite common, to best understand weakness and model in real-world scenario, the community is heading towards more complex

36th Conference on Neural Information Processing Systems (NeurIPS 2022)., New Orleans.

evaluations involving fine-grained evaluation [61] across several metrics (or criteria [46, 103]) and several tasks [36, 47, 63, 89, 91, 108]. This is due to the increasing performance of deep neural networks, which are nowadays designed to generalize in a great variety of situations and to solve complex tasks [86]. One is typically seeking for models with good *transfer learning* properties, meaning an ability to generalize well under distribution shift and/or task shift [51].

**How to aggregate performances?** The multi-tasks setting has been investigated in recent works that provide benchmark of state-of-the-art models across a great variety of tasks [28, 62, 80, 90, 108], sometimes with more than fifty [2, 84, 85, 94]. These papers provide tables of scores across the considered tasks, but the only non-qualitative way to compare systems consists in averaging the performances across tasks and then ranking systems according to their mean score values. This is, for instance, done with the GLUE benchmark [91] and its derivatives [92]. However, taking the mean is seriously flawed since the different metrics are usually not on the same scales and can even be unbounded [23, 102]. Even a pre-processing renormalization scheme would fail to capture the intrinsic difficulty of the tasks.

**Contribution 1.** Our first contribution is to provide a reliable tool to rank systems in a multi-tasks setting. We rely on a ranking aggregation procedure which, from a set of rankings induced by each criterion, returns a single ranking that somehow aggregates the former. This procedure, called the *Kemeny consensus* [52], can be seen as a voting rule and stems from the social choice theory [66].

**Aggregation when instance-level information is available.** As illustrated by Ruder [83], Zhong et al. [109], a fine-grained understanding of the model performance should include instance-level scores. If taking the mean is quite natural in the classification setting, this is not always the case, as recently pointed out by [73] in the NLG setting. In this article, the authors investigate pairwise comparison of NLG systems for a single metric (*e.g.* BLEU [71], ROUGE [59], METEOR [5, 35, 49], CHRF [76, 77], BertScore [105]). They prove that a comparison based on the mean or the median of the scores across test utterances can be highly flawed. They rather advise to rely on the Bradley-Terry [10] pairwise comparison method, which consists, for two systems A and B, in computing the proportion of utterances on which A achieves a better score than B. Their work is a significant advance but remains limited to pairwise comparisons.

**Contribution 2.** Our second contribution consists in going one step further than [73] by applying our ranking procedure to an arbitrarily large set of NLG systems with respect to a group of fixed criterion. Our evaluation methodology can be seen as a natural extension of [73] since it coincides with the latter in the particular case of pairwise comparison. In a more realistic multi-criteria scenario, we combine our two contributions and develop a *two-stages ranking aggregation procedure* which first aggregates along utterances and then along criteria.

**Experiments.** Our two contributions rely on our aggregation procedure which is proved to be effective through several experiments.

1. We explain on a simple synthetic example the superiority of our approach compared to the mean-aggregation procedure and the pairwise-aggregation procedure, both in terms of consistency and robustness.

2. We use our ranking procedure on 10 multi-tasks / multi-criteria benchmarks and observe it leads to different conclusions than mean- and pairwise-aggregation procedures.

3. We argue our procedure is more robust by investigating its stability with respect to the addition of criteria and with respect to the addition of systems.

Our code and the collected data will be released to accelerate the adoption of what we think is a reliable evaluation method for multi-tasks and multi-criteria benchmarks.

## 2 Problem Formulation and limitations of existing methods

### 2.1 General Considerations

When comparing systems performances, two settings can be distinguished depending on the information granularity at our disposal and on the way one wishes to use this information. In general, each system is scored on each instance of a test set with respect to a given metric. The final (single) score of the system with respect to this metric is obtained through an aggregation procedure we

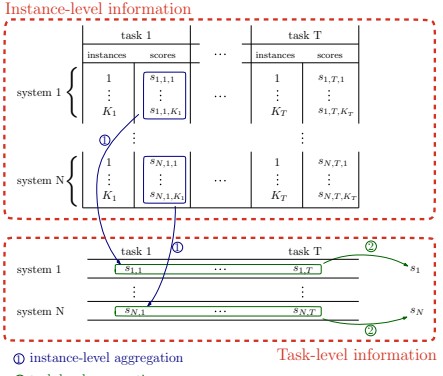

Figure 1: Illustration of the two considered frameworks relying on different information granularity: task-level information (below) or instance-level information(above). From the latter, one can derive the former relying on what we call an instance-level aggregation. A task-level aggregation can then be performed to synthesize a system performance.

will call *instance-level aggregation*, and which has to be chosen by the practitioner (usually, the mean of the instances-scores). Then, the final benchmark score of the system is obtained through an aggregation we call *task-level aggregation* of the scores of this system for each metric of the considered benchmark. See Fig. 1 for an illustration.

**Notations.** Suppose we are given $N \geq 1$ systems evaluated on $T \geq 1$ tasks, each task $t \in \{1, \dots, T\}$ being associated with a metric and a test set made of $K_t \geq 1$ instances. For every $n \in \{1, \dots, N\}$, $t \in \{1, \dots, T\}$ and $k \in \{1, \dots, K_t\}$, we denote by $s_{n,t,k} \in \mathbb{R}$ the score of system $n$ on the instance $k$ of task $t$.

**Instance-level aggregation.** The performance of system $n$ on task $t$ is an aggregation of its scores $(s_{n,t,k})_{1 \leq k \leq K_t}$ on each instances. This aggregation is chosen by the practitioner and is usually the *mean-aggregation* defined by $s_{n,t}^{\text{mean}} := \frac{1}{K_t} \sum_{1 \leq k \leq K_t} s_{n,t,k}$.

**Task-level aggregation.** Sometimes, one only has access to the aggregated scores of each system on each task, that is, for every $n \in \{1, \dots, N\}$ and $t \in \{1, \dots, T\}$, to a score $s_{n,t} \in \mathbb{R}$ which corresponds to an instance-level aggregation for system $n$ on the instances of task $t$. From these quantities, one can then compute, for each system, a synthetic score reflecting the overall performance of this system across every considered task. Again, the usually-taken synthetic score of system $n$ is the mean-aggregation: $s_n^{\text{mean}} := \frac{1}{T} \sum_{1 \leq t \leq T} s_{n,t}$.

## 2.2 Problem Formulation

**Ranking objective.** When benchmarking systems, the goal is to output a ranking of each systems according to some objective criterion. Formally, we need to introduce the symmetric group on $N$ elements, denoted by $\mathfrak{S}_N$, which elements are the $N!$ permutations of $\{1, \dots, N\}$. Equipped with this notation, our goal is to output a permutation

$$\sigma^* = [\sigma_1^*, \dots, \sigma_N^*] \in \mathfrak{S}_N,$$

corresponding to the rankings of the systems. For instance, one reads "system $i$ is the $\sigma_i$-th best system". Depending on the granularity of the information at our disposal, we distinguish two problems.

**Ranking Systems from Task Level Information.** *Given a set of scores $(s_{n,t}, 1 \leq n \leq N, 1 \leq t \leq T)$ of $N$ systems on $T$ tasks, find a proper aggregation procedure.*

**Ranking Systems from Instance Level Information.** *Given a set of scores $(s_{n,t,k}, 1 \leq n \leq N, 1 \leq t \leq T, 1 \leq k \leq K_t)$ of $N$ systems on the different instances of $T$ tasks, find a proper aggregation procedure.*

## 2.3 Limitation of Existing Methods

**Mean-aggregation procedure.** The mean-aggregation procedure consists in taking the permutation $\sigma_*^{\text{mean}}$ that would rank the aggregated means $s_n^{\text{mean}}, 1 \leq n \leq N$. This procedure suffers from several flaws. First, it is not well-suited when the metrics associated with the considered tasks are not on the same scale. Consider, for instance, the situation where one of the tasks (say task $t_0$) is associated with a metric that is at a significantly larger scale than the others. In that case, the ranking obtained through mean-aggregation would probably correspond to the ranking induced by task $t_0$. One could argue that a remedy would first normalize each metric so that everything is on the same scale. However, the

|   | Task1 | Task2 | Task3 | Task4 | Task5 | Task6 | SUM |
|---|-------|-------|-------|-------|-------|-------|-----|
| A | 0.3 (3) | 5 (3) | 10 (1) | 0.02 (2) | 1.0 (1) | 0.4 (3) | 16.72 (13) |
| B | 0.1 (2) | 4 (2) | 13 (2) | 0.01 (1) | 2.2 (3) | 0.3 (2) | 19.61 (12) |
| C | 0.0 (1) | 3 (1) | 15 (3) | 0.03 (3) | 2.0 (2) | 0.2 (1) | 20.23 (11) |

Table 1: Example of where pairwise rankings can be paradoxical. Mean aggregation outputs $A > B > C$ while pairwise ranking considered in [73] fails to rank the systems and produce $B > A, C > B, A = C$. Our method does not have this flaw and outputs $C > B > A$.

resulting aggregation would still fail to capture each task's intrinsic difficulty. Worse, this procedure is impractical in cases where some metrics are unbounded – for instance, this is the case of the BARTScore [102]. Finally, another weakness of the mean-aggregation ranking procedure is that the score of a system is computed irrespective of its relative performance with respect to the others. This simple observation has been pointed out by [73] who advises, in the special case of two systems and one metric, to compute the number of times a system is better than the other on the instances.

**Pairwise ranking.** To be a bit more formal, the pairwise ranking aggregation proposed by [73] to rank two systems $A$ and $B$, which scores on a given task are given by $s_1^A, \ldots, s_K^A$ and $s_1^B, \ldots, s_K^B$, consists in computing

$$\lambda_A := \sum_{k=1}^{K} \mathbf{1}_{s_k^A \geq s_k^B} \quad \text{and} \quad \lambda_B = K - \lambda_A.$$

Then, $A$ is better than $B$ if and only if $\lambda_A > \lambda_B$. As explained by the authors, this method is more relevant than the mean-aggregation in the context of NLG evaluation. However, it is limited to the evaluation of two systems and does not apply for a general number $N \geq 3$ of systems. A solution would be to give a rank for each pair of systems and then aggregate these pairwise rankings. However, this would lead to a prohibitive $\binom{N}{2}$ computational factor for the complexity. Moreover, the conclusion of these pairwise rankings can be paradoxical. Tab. 1 below provides a toy example where three systems $A$, $B$, and $C$ are evaluated on 6 tasks, and where the pairwise comparisons give the paradoxical conclusion $B > A, C > B$ and $A = C$.

# 3 Ranking via Kemeny consensus

We now turn to the description of our methodology to rank an arbitrary number of systems on multi-tasks / multi-criteria benchmarks.

## 3.1 Kemeny Consensus

Let us consider the problem of ranking $N$ systems on $T$ tasks based on the information of the scores $s_{n,t}$ of each system $n$ on each task $t$. We believe a robust approach to this problem consist in relying on the relative performance between systems on each task. More precisely, for each task $t \in \{1, \ldots, T\}$, we consider

$$\sigma^t = [\sigma_1^t, \ldots, \sigma_N^t] \in \mathfrak{S}_N,$$

where $\sigma_n^t$ corresponds to the rank of system $n$ on task $t$, in decreasing order. Then, we would like to find an appropriate procedure that aggregates the $T$ rankings $\sigma^1, \ldots, \sigma^T$. More formally, we would like to define a function

$$f : \underbrace{\mathfrak{S}_N \times \cdots \times \mathfrak{S}_N}_{T \text{ times}} \longrightarrow \mathfrak{S}_N.$$

This function, from a set of $T$ permutations corresponding to rankings, should return a final permutation that summarizes them. One difficulty is that the mean procedure makes no sense on the symmetric group, which is not a vector space. It turns out that a very natural choice consists in taking $f$ as the so-called *Kemeny consensus* [52] aggregation procedure, which somehow corresponds to compute a barycenter.

**Kemeny consensus.** *Let $d$ be the Kendall distance on the symmetric group, defined for every $\eta, \tau \in \mathfrak{S}_N$ by $d(\eta, \tau) := \sum_{1 \leq i, j \leq N} \mathbf{1}_{(\eta_i - \eta_j)(\tau_i - \tau_j) < 0}$. A Kemeny consensus $\sigma^*$ of $\sigma^1, \ldots, \sigma^T$ is a solution of the following minimization problem $\min_{\sigma \in \mathfrak{S}_N} \sum_{1 \leq t \leq T} d(\sigma^t, \sigma)$.*

**Why is Kemeny consensus natural?** As proved by Young and Levenglick [100], the Kemeny consensus aggregation procedure is the only rule that satisfies three natural properties: *neutrality*,

meaning that it does not depend on the order of the tasks; *consistency*, meaning that if the tasks are split in two subsets and that the aggregation in the to subsets rank system $i$ above system $j$ then $\sigma_i^* > \sigma_j^*$ ; and *the Condorcet criterion* [27], meaning that an item wining all its pairwise comparison is ranked fist. Moreover, the Kemeny consensus is also the maximum likelihood of the widely-used *Mallows* statistical on the symmetric group [99].

## 3.2 Borda's count approximation

If the Kemeny consensus is the ideal objective one would like to obtain, its computation is, in general, an NP-hard problem [6, 40] – although some regularity assumptions, rarely satisfied in practice, can speed up the computation, see for instance [8] and [11]. Fortunately, there exist many ways to get satisfying approximations of the latter: see for example [1] for a comprehensive empirical study. For our experiments, we choose the so-called Borda's count procedure, defined hereafter for the instance-level and/or task-level aggregation.

**Borda's count.** *The Borda's count consists, from a set of permutations $\eta^1, \ldots, \eta^L \in \mathfrak{N}$ corresponding to the ranking of $N$ systems across $L \geq 1$ tasks or instances, to sum the ranks of each system and then to rank the obtained sums. Formally, it*

1. *Compute* $\mathrm{sum}_n := \sum_{l=1}^{L} \eta_n^l$ *for every* $1 \leq n \leq N$,

2. *Output* $\eta := \mathrm{Borda}(\eta^1, \ldots, \eta^L) \in \mathfrak{S}_N$ *that ranks the sums,* $\mathrm{sum}_n$ *(*$\mathrm{argsort}(\mathrm{argsort}(\mathrm{sum}_1, \ldots, \mathrm{sum}_T))$*)*.

There are at least four explanations for choosing Borda's count procedure. First, it coincides with the pairwise ranking procedure in the case of two systems, making it a natural generalization. Second, there exists a theoretical result assessing it is a 5-approximation of the Kemeny consensus [30] with respect to the Kendall distance [42]. Third, it is an unbiased estimator of the Kemeny consensus with low sample complexity for data distributed according to standard rankings models such as Mallows [13, 44]. Fourth, from a practical perspective, [1] observe it is efficient, accurate, and actually 10 times faster than the other approximation algorithms. Fourth, We are now in a position to give our answers to the initial ranking problems from Task Level Information and from Instance Level Information.

## 3.3 Our Ranking Procedures

**How to rank Systems from Task Level Information.** *Let $\sigma^t \in \mathfrak{S}_N$ be permutation that ranks the scores $s_{1,t}, \ldots, s_{N,t}$. Our aggregation procedure ($\sigma^*$) output is $\sigma^* := \mathrm{Borda}(\sigma^1, \ldots, \sigma^t)$.*

**How to rank Systems from Instance Level Information.** *We actually give two different procedures. For every task $t \in \{1, \ldots, T\}$ and every instance $k \in \{1, \ldots, K_t\}$ of that task, let $\sigma^{t,k}$ be the permutation that ranks the scores $s_{1,t,k}, \ldots, s_{N,t,k}$. See Figure 2 for an illustration.*

**Two-level aggregation ($\sigma^{2l}$).** *This procedure*
1. *Compute $\sigma^t := \mathrm{Borda}(\sigma^{t,k}, 1 \leq k \leq K_t)$ for each task $t \in \{1, \ldots, T\}$,*
2. *Output $\sigma^{2l} := \mathrm{Borda}(\sigma^t, 1 \leq t \leq T)$.*

**One-level aggregation ($\sigma^l$).** *This procedure outputs $\sigma^l := \mathrm{Borda}(\sigma^{t,k}, 1 \leq t \leq T, 1 \leq k \leq K_t)$.*

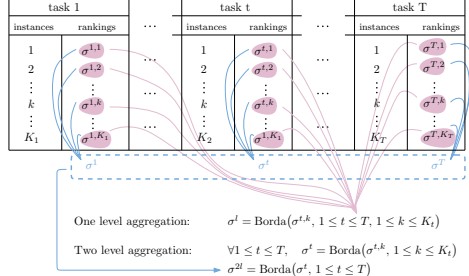

Figure 2: Illustration of our two aggregation procedures to rank systems from instance-level information.

## 3.4 How to compare rankings

The rest of the paper is dedicated to synthetic and empirical experiments, on which we demonstrate the soundness of our approach. In order to obtain a quantitative result, one needs to be able to compare different rankings quantitatively. Two measures can be used for that purpose: (1) the Kendall

distance and (2) the Kendall correlation ($\tau$) [53, 54]. The Kendall distance computes the number of inversions between two permutations and is therefore adapted for our purpose of ranking systems. The values of $\tau$ range from $[-1, 1]$ where the value of 1 corresponds to a strong agreement, and close to -1 indicates strong disagreement.

## 4 Synthetic Experiments

In this section, we validate on simulated data the performance of our method on two criteria: robustness to manipulation and robustness to scaling.

### 4.1 Data Generation

The toy experiment analysis is carried out on synthetic scores over $N = 20$ systems, $T = 20$ tasks and $K = 20$ instances. For each $n \in \{1, \ldots, N\}$, we model the performance of system $n$ by a Gumbel r.v. $G_n$ centered at $\phi * n$ at scale $\beta = 1$, where $\phi \in [0, 1]$ is a dispersion parameter. The scores of system $n$, $(s_{n,t,k})_{t,k}$, are i.i.d. samples of $G_n$ centered at $\phi * n$ with scale $\beta = 1$, where $\phi \in [0, 1]$ is a dispersion parameter. Moreover, the scores of different systems are sampled independently. Since $G_{n+1} - G_n$ follows a logistic distribution with mean $\phi$ at scale 1, this imply that $\mathbb{P}(G_{n+1} - G_n > 0) > 0.5$, the probability that system $n + 1$ performs better than system $n$ is at least $0.5$. Therefore, for all $k, t$, the rankings of systems is a realization of the *ground-truth ranking* $[1, ...N]$, with a noise term controlled by the 'dispersion' parameter $\phi$.

Extreme scenarii correspond to the choices $\phi = 0$ and $\phi = 1$. More precisely, $\phi = 0$ implies that all scores $s_{n,t,k}$ have the same distribution, whereas $\phi = 1$ induces a strong consensus, *i.e.*, a clear system ranking emerges.

**Remark 1.** *Sampling $s_{n,t,k}$ according to the described procedure is equivalent to sampling the ranking of the systems from the well-know Plackett-Luce distribution [39, 75] with weights $w_i = \phi * n$. Interestingly, this distribution over ranking can be seen both from the utilitarian perspective in which the scores $s_{n,t,k}$ are real numbers and from a ranking-model perspective in which the ranking of systems have known distribution.*

### 4.2 Robustness to manipulation

**Setting.** To test the robustness of our ranking procedure, we analyze its stability with respect to perturbations of the scores. More precisely, our way to corrupt scores of a given task $t$ consists in sampling $(sn, t, k)_{n,k}$ as i.i.d. samples following Gumbel distribution centered at $-n$. This implies that, for that task $t$, the underlying ranking is $[N, ...1]$, namely the exact opposite of the ground truth $[1, \ldots, N]$. The robustness to manipulation analysis shows how the error on the final ranking of systems increases as the scores of some $t$ tasks are 'corrupted'. Here, the error is computed relying on the normalized Kendall distance between the ground-truth ranking $[1, ...N]$ and the ranking of systems obtained relying on the corrupted scores.

**Results:** For each of the considered methods, $\sigma^{mean}$, $\sigma^l$ and $\sigma^{2l}$ we report in Fig. 3 the results of the robustness analysis when $\phi$ and the number of corrupted task varies. The results of the robustness analysis show that $\sigma^{2l}$ outperforms $\sigma^l$ which at the same time consistently outperforms $\sigma^{mean}$. Overall, for the same number of corrupted tasks and dispersion, the error of $\sigma^{2l}$ is always the smallest. Moreover, the score-based method $\sigma^{mean}$ gets an error larger than .75 when just 2, 3, and 5 out of the total of $T = 20$ tasks have been corrupted, while for $\sigma^l$ the same error is achieved with 5, 7 and 10 corrupted tasks. The most robust method is the two-level $\sigma^{2l}$ for which 10, 11 and 11 out of $T = 20$ tasks have to be corrupted to get the same error of 0.75.

**Takeaways:** We conclude that the ranking-based methods are more robust than $\sigma^{mean}$. In particular, the 2-level aggregation $\sigma^{2l}$ is the most robust aggregation procedure.

### 4.3 Robustness to scaling

To further compare the ranking, we corrupt the scores of a given task by re-scaling them by a factor of $x > 0$. Whereas it does not affect our ranking procedure (every ranking induced by a task-instance pair remains the same), it increasingly perturbs the mean aggregation procedure as $x$ increases. Re-scaling the scores by a factor of 2 produces an error larger than 90%. For larger $\phi$, re-scaling the

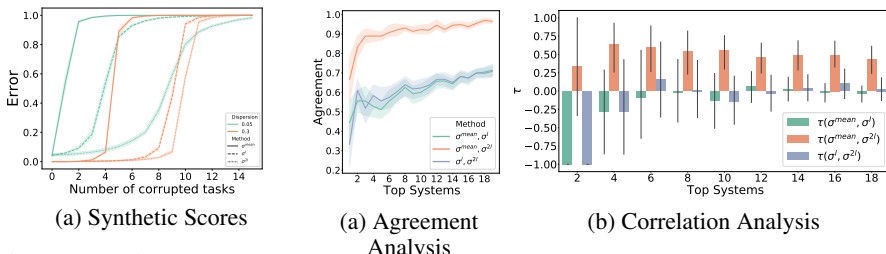

(a) Synthetic Scores

Figure 3: Robustness on synthetic scores.

(a) Agreement Analysis

(b) Correlation Analysis

Figure 4: Global Analysis of Instance Level Raning

scores by a factor of 7 produces the same error (see Fig. 7 for detailed results).
**Takeaways:** Re-scaling one task's score with an arbitrarily large number will always produce an arbitrarily large error for mean aggregation while not affecting ranking based aggregation.

## 5 Empirical Experiments

In this section, we present our results on real evaluation scores. Our large scale experiments relies on real evaluation scores (over 270k scores) which are described in Ssec. 5.1. In Ssec. 5.2 we gather experimental results for *Ranking Systems from Task Level Information*, while Ssec. 5.3 is dedicated to the problem of *Ranking Systems from Instance Level Information*.

### 5.1 Data Collection

**Datasets with Task Level Information** We collect the results of GLUE [91], SGLUE [92][1] and XTREME [50]. For GLUE the dataset is composed of $N = 105$ systems that are evaluated on 9 different tasks: CoLA [93], SST-2 [87], MRPC [38], STS-B [14], QQP, MNLI [96], QNLI [80], RTE [7, 31, 48] and WNLI [57]. For SGLUE, the final dataset gathers scores from $N = 24$ systems that are evaluated on 10 different tasks: BoolQ [17], CB [33], COPA [82], MultiRC [55], ReCoRD [104], RTE, WiC [74], WSC and its derivatives AX-b AX-g [57]. XTREM benchmark is composed of $N = 15$ systems and include tasks such as sentence classification (using XNLI [29, 95] and PAXS-X [98, 106]), structured prediction (relying on Universal Dependencies v2.5 [68] and Wikiann [70, 79]), sentence retrieval (with BUCC [110, 111] and Tatoeba [3]) and question answering (via XQuAD [4, 80], MLQA [58], TyDiQA-GoldP [18]). For all benchmarks, various types of metrics with various scales are reported (*i.e* accuracy, f1, correlation).

**Datasets with Instance-level information** In this setting we focus on NLG evaluation as these scores are among the easiest to be collected. We focus on five different tasks: summary evaluation, image description, dialogue and translation. For *summary evaluation*, we use TAC08 [32], TAC10, TAC11 [69], RSUM [9] and SEVAL [41]. For *sentence-based image description* we rely on FLICKR [101] and for *dialogue* we use PersonaChat (PC) and TopicalChat (TC) [64]. Finally for *machine translation*, we rely on the multilingual quality estimation (MLQE) introduced in Ranasinghe et al. [81]. For all datasets except MLQE, we consider automatic metric based on S3 (both variant pyr/resp) [72], ROUGE [59] (including 5 of its variants [67]), JS [1-2] [60], Chrfpp [77], BLEU, BERTScore [105], MoverScore [107]. For MLQE we solely consider several version of BERTScore, MoverScore and ContrastScore. We also add human evaluation which is specific to each dataset. All details corresponding to these dataset can be found in Appendix A.

### 5.2 Task-level Aggregation Experiments

In this section, we address the aggregation problem when task-level information is available. We first study the final ranking obtained by different methods on GLUE, SGLUE, and XTREM. Then, we assess the robustness of $\sigma^*$ when removing tasks.

**Comparison with mean-aggregation** To compare the rankings $\sigma^{mean}$ and $\sigma^*$, we compute *(i)* the

---

[1]Results can be found at https://super.gluebenchmark.com/

| Dataset | Top 1 | Top 3 | Top 5 | Top 10 |
|---------|-------|-------|-------|--------|
| XT. | 1 | 0.66 | 0.8 | 0.9 |
| GLUE | 1 | 1 | 0.8 | 0.8 |
| SGLUE | 1 | 1 | 0.8 | 0.9 |
| Dataset | Last 3 | Last 5 | Last 10 | $\tau$ |
| EXT. | 1 | 0.8 | 0.9 | 0.82 |
| GLUE | 1 | 0.8 | 0.7 | 0.92 |
| SGLUE | 1 | 1 | 1 | 0.91 |

Table 2: Agreement count between Top N/Last N systems on the Ranking when Task Level Information is available. $\tau$ is computed on the total ranking.

| GLUE | | | XTREM | | |
|------|------|------------------|------|------|------------------|
| $\sigma^*$ | Team | $\sigma^{mean}$ | $\sigma^*$ | Team | $\sigma^{mean}$ |
| 0 (1430) | Ms Alex | 0 (88.6) | 0 (55) | ULR | 0 (83.2) |
| 1 (1405) | ERNIE | 1 (88.0) | 1 (50) | CoFe | 1 (82.6) |
| 2 (1397) | DEBERTA | 2 (87.9) | 2 (44) | InfoLXL | 3 (80.6) |
| 3 (1391) | AliceMind | 3 (87.8) | 3 (42) | VECO | 4 (80.3) |
| 4 (1375) | PING-AH | 5 (87.6) | 4 (35) | Unicoder | 5 (79.4) |
| 5 (1362) | HFL | 4 (87.7) | 5 (34) | PolyGlot | 2 (80.6) |
| 6 (1361) | T5 | 6 (87.5) | 6 (31) | ULR-v2 | 6 (79.4) |
| 7 (1358) | DIRL | 10 (86.7) | 7 (29) | HiCTL | 8 (79.1) |
| 8 (1331) | Zihan | 7 (87.6) | 8 (29) | Ernie | 7 (79.1) |
| 9 (1316) | ELECTRA | 11 (86.7) | 9 (21) | Anony | 10 (78.3) |

Table 3: Qualitative analysis between ranking obtained with $\sigma^*$ or $\sigma^{mean}$. Results in parenthesis report the score of the considered aggregation procedure.

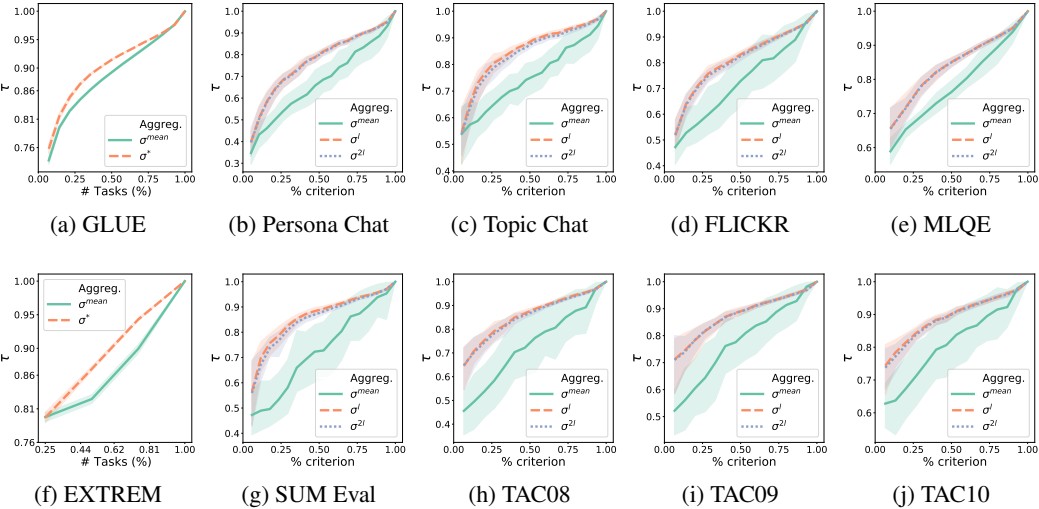

(a) GLUE   (b) Persona Chat   (c) Topic Chat   (d) FLICKR   (e) MLQE

(f) EXTREM   (g) SUM Eval   (h) TAC08   (i) TAC09   (j) TAC10

Figure 5: Impact of adding/removing metrics/tasks. The first column refers to ranking obtained with task-level information, while others columns refer to ranking obtained with instance-level information.

agreement rate (in %) which is the proportion of common top-ranked systems between $\sigma^{mean}$ and $\sigma^*$, and *(ii)* the Kendall Tau correlation ($\tau$) between the rankings.

**Results.** In Tab. 2, we compare the rankings of aforementioned methods for Top K systems (strongest systems) and Last K systems (weakest systems). For the three benchmarks, we observe a high correlation between the final rankings (*i.e.* correlation values are in the range $[0.80, 1]$). To a finer degree, we also observe that methods tend to agree on which are the best/worst systems. Although $\sigma^{mean}$ and $\sigma^*$ agree on the best/worst systems, they do not rank them in the same order (see Tab. 3). For instance, on XTREM, the third-best system according to $\sigma^{mean}$ (rank 2) is actually the sixth-best system according to $\sigma^*$.

**Takeaways.** When changing the aggregation function, the response to our initial question "what are the best systems?" varies.

**How does the addition/removal of new tasks/metrics affect the ranking?** When building a benchmark, practitioners can always add new tasks to refine the model performance assessment (it boils down to adding a new column in Tab. 1). In this experiment, we analyze how adding and removing tasks affect the rankings of the aggregation procedures.

**Setting.** We compare the rankings obtained when considering a subset of the tasks and the one obtained using all the tasks. Formally, for a given number of tasks $t \leq T$, we randomly sample $t$ tasks, compute the rankings obtain by our procedure and by the mean procedure, $\sigma^{*,t}$ and $\sigma^{mean}$, on these $t$ tasks, and finally compute the Kendall correlation between $\sigma^{*,t}$ (resp. $\sigma^{mean,t}$) and the "ground truth" $\sigma^*$ (resp. $\sigma^{mean}$). We repeat this random sampling 100 times to obtain a mean/variance plot of the correlations in the function of the number of sub-tasks.

**Results.** We report in Fig. 5(a,f) the obtained results for varying size of subsets. Interestingly we

observe that correlation between the $\sigma^{*,t}$ and $\sigma^*$ is consistently higher than the one between $\sigma^{mean,t}$ and $\sigma^{mean}$. This difference is particularly visible in the range $[0.25\% - 0.75\%]$. We observe a similar behavior when considering SGLUE (see Fig. 8)

**Takeaways.** The ranking from $\sigma^*$ is more robust to task addition/drop than the one from $\sigma^{mean}$.

## 5.3 Instance-level Aggregation Experiments

For instance level aggregation, we conduct experiments on the 9 aforementioned data-sets. We study both the final ranking obtained for each aggregation (*i.e.* $\sigma^l$, $\sigma^{2l}$ and $\sigma^{mean}$) as well as the effect of task addition/removal.

**Global analysis** When conducting our experiments, we observe that the three different aggregation procedures lead to three different state-of-the-art in 8 datasets out of 9. Furthermore, they never agree on the top 3 systems. In what follows, we compare $\sigma^l$, $\sigma^{2l}$ and $\sigma^{mean}$.

**Setting.** We compare the obtained ranking by (1) comparing the Kendall correlation (see Fig. 6), (2) comparing the number of agreements between the top N systems, (3) computing the Kendall correlation between them.

|  | PC | TC | FLI. | MLQE |
|---|---|---|---|---|
| $\tau(\sigma^l, \sigma^{2l})$ | -0.08 | -0.01 | 0 | -0.03 |
| $\tau(\sigma^{mean}, \sigma^{2l})$ | 0.32 | 0.27 | 0.29 | 0.01 |
| $\tau(\sigma^{mean}, \sigma^l)$ | -0.10 | -0.15 | -0.04 | 0.00 |
| RSUM | SEVAL | TAC08 | TAC09 | TAC11 |
| 0.04 | 0.14 | 0.28 | 0.06 | -0.06 |
| 0.07 | 0.52 | 0.32 | 0.37 | 0.37 |
| 0 | 0.10 | 0.23 | 0.19 | 0.07 |

Figure 6: $\tau$ on global instance-level rankings.

**Results.** When considering the agreement analysis of Fig. 4(a), we observe that $\sigma^{2l}$ and $\sigma^{mean}$ select a high number of common top systems. However, the correlation of the rankings induced by $\sigma^{2l}$ and $\sigma^{mean}$ on these top-systems is low (see Fig. 4(b)). This is also the case for the correlation of the entire rankings (see Fig. 4). In short $\sigma^{2l}$ and $\sigma^{mean}$ select similar systems but rank them differently. Similar analysis shows that $\sigma^l$ disagrees from $\sigma^{2l}$ and $\sigma^{mean}$ both on top systems and on their orders.

**Takeaways.** $\sigma^{2l}$ exhibits a more similar behavior than $\sigma^l$ with respect to $\sigma^{mean}$.

**What is the impact of removing/adding tasks?** In NLG, different metrics (*i.e.* task) assess the quality of a generated sentence along a different axis. As adding a new metric may affect the final ranking, we investigate the impact of task addition/removal on the final system ordering.

**Setting.** Similarly to **??**, we study the evolution between the correlation between the ranking computed on a subset of tasks and the ground truth ranking (computed on all tasks) for each of the three procedures.

**Results.** We observe that for all datasets both $\sigma^{2l}, \sigma^l$ obtain higher correlation and lower variance compared to $\sigma^{mean}$ when adding/removing tasks. Results for RSUM reports a similar trends (see Fig. 9).

**Takeaways.** The ranking obtained with either $\sigma^l$ or $\sigma^{2l}$ are more robust to task addition/drop than the one from $\sigma^{mean}$.

## 6 Conclusion and Future Works

In this paper, we introduced new aggregation procedures to rank systems when either task level scores or instance level scores are available. Our methods, which are theoretically grounded and rely on Kemeny ranking consensus, address fundamental flaws of the widely used arithmetic mean.

We conducted extensive numerical experiments, which show that our methods are both more reliable and more robust than the mean aggregation while leading to different conclusions on which are the best systems. Overall, when task-level (resp. instance-level) information is available, we would recommend using the aggregation procedure $\sigma^*$ (resp. $\sigma^{2l}$) rather than the $\sigma^{mean}$ (resp. $\sigma^{mean}$ and $\sigma^l$).

Although we focused on NLP benchmarks, our methodology could be applied in other modalities (*e.g.* Computer Vision, Audio). Another interesting avenue would be to consider a weighted version of the Kemeney consensus where weight could reflect task-specific criteria (*e.g* fairness, robustness, complexity).

# 7 Acknowledgements and Fundings

We thank Maxime Peyrard, Saibo Geng, and Wei Zhao for sharing their collected dataset. A warm thanks to Matthieu Labeau for his sharp eyes and excellent comments when reading our paper manuscript. This work was granted access to the HPC resources of IDRIS under the allocation 2021-101838 and 2021-101913 made by GENCI. Nathan is funded by the projet ANR LIMPID.

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
