# OpenReview forum: "What are the best Systems? New Perspectives on NLP Benchmarking"
_NeurIPS.cc/2022/Conference — NeurIPS 2022 Accept_

### Official Review · Reviewer_qERs · 2022-07-12

**Rating:** 6
**Confidence:** 3
**Soundness:** 3 good
**Presentation:** 3 good
**Contribution:** 3 good

**Summary:**

This paper proposes a method for ranking a set of systems based on their performance across multiple tasks. The authors argue that the naive method of averaging task scores can be biased if the scores have different ranges. They advocate ranking in order of summed per-task ranks instead (Borda's count procedure), citing theoretical support for this approach. A variant "two-level" algorithm, applicable when instance-level scores are available, first applies the Borda procedure to combine per-instance system rankings within each task, then applies the procedure again to combine the resulting per-task rankings.

These methods are compared to ranking using mean scores on both synthetic and real data. On synthetic data, the two-level approach is found to be most resistant to scoring noise. On a large number of NLP tasks, both proposed methods (standard and two-level) are more robust to adding or dropping tasks. They assign global rankings to systems that are similar to the baseline - putting the same systems near the top or bottom - but make fine-grained adjustments, sometimes resulting in different overall winners.


**Questions:**

43: This implies that your method does capture task intrinsic difficulty, but I can’t see that this is the case.

133: I don’t really buy the complexity argument. Even if there are O(100) systems, there are still only O(10^4) pairs, which seems completely manageable.

149: If you are willing to relax the interpretation of ranks as permutations, and think of them as scores, then taking the mean rank actually seems quite intuitive. It’s basically the same as the Borda approximation, except that means contain more information than ranks, e.g. systems whose average ranks are 1.1 and 1.2 are closer than would be implied by the Borda ranks of 1, 2.

Figure 2: typo in the “task t” column: K_2 -> K_t, K_1 -> K_t

3.3 This description is a bit confusing. I imagine both schemes produce a per-task ranking that gets combined using Borda, with the difference being that in one-level the per-task ranking comes from mean scores across instances, whereas in two-level the per-task ranking comes from applying Borda over instances. If that’s the case, consider adding a bit of text like the above, or making the notation and diagram more expressive.

Table 3: What are the scores in the sigma* column? If these are the Borda rank sums, I would expect the top-ranked systems to have smaller values.

329: Missing reference.

**Limitations:**

Yes.

**Strengths And Weaknesses:**

Strengths:

- Addresses an important problem, and proposes a simple, well-motivated solution that performs well. The idea isn’t novel, but it isn’t well known in NLP, and I think it has the potential to be widely adopted.

- Does a good job motivating the proposal and citing extensive relevant literature.

- Clearly organized and well presented, with a few exceptions detailed below.

- Carries out experiments on a broad range of task settings.

Weaknesses:

- A bit thin on substance. A lot of the paper is taken up with a formal presentation of the method, which is quite easy to grasp intuitively. Some more space might have been devoted to analysis of the results, to provide more evidence that the new rankings are better (a difficult problem, however).

- An obvious objection to summarizing scores with ranks is that it throws away information about how close scores are to each other. One way to deal with this is to allow ties, for instance in cases where there isn’t statistical support for assigning different ranks. The whole aspect of confidence intervals on the scores is completely missing from the paper.

---

> ### Author Response · Authors · 2022-08-01
> **Reply to reviewer qERs**
>
> We would like to warmly thank reviewer qERs for carefully reading our manuscript and for their enthusiasm about our work. We indeed hope that our work will be widely adopted by the community as we firmly believe it provides a more robust way to evaluate NLP systems.
>
> Here are our responses to qERs concerns.
> * We agree that the experimental part is quite compressed as we thought it was very important to make the problem statement and our solution crystal clear. The additional page of the eventual camera-ready version will allow us to provide more details on the experiments and their analysis. We also refer you to the supplementary materials which contain many more experiments. Concerning your side remark about the way to say a ranking is better than another, it is indeed a quite complicated question. Our view on this is to say that, the more robust a ranking aggregation is, the more confident practitioners should be on its outcomes. From this standpoint, our experiments prove our method is better.
>
> * We agree with your point which was already raised by the two other reviewers. To alleviate this variability issue, we propose to introduce two new functionalities to our method. Even if not directly related, we also refer reviewer qERs to our supplementary material where some variability computations are performed (see appendix C).
>      * First, for a given task, we will compute the variability of the score and then, if two systems have scores whose difference is below the computed variability, they will be considered as equally performant. This will allow for ties and the Borda count will be performed the same way, except its outcome will also allow for ties.
>
>      * Second, in order to allow practitioners to take into account intrinsic difficulty of tasks (which has also to do with variability), we will incorporate a weighted version of our procedure. Each task t will be (to the user’s discretion) associated to a weight $w_t$ and the weighted Borda count will be $\sum_t w_t \eta_n^t$ for system n.
>
> Let us now turn to the more specific questions of reviewer qERs:
> * l.43. As mentioned above, we will add a weighted version of the Borda count which will properly allow us to handle intrinsic tasks difficulties.
> * l.133. This is more of an abstract computational argument. However it could very well happen that in practice, several models let say are tested on several checkpoints and random seeds, which would quickly bring us to infeasible computations.
> * l.149. Actually, Borda count is computing the mean rank up to a normalization term. Since we are interested in the final ranking of the systems, averaging or not would lead to the same result. However, we agree that providing the renormalized Borda count can also provide practitioners information about the relative performances of systems, in addition to the final ranking. We will add this remark to the manuscript, thanks!
> * We will correct the typo in Fig 2.
> * Actually, the one-level ranking aggregation does not provide a ranking per task, it directly outputs a final ranking. Let us recall the procedure here. Each pair (t,k) of task t and utterance k leads to a ranking $\sigma_{t,k}$ of the N systems. In the one-level aggregation procedure, we directly compute the Borda aggregation of all of these \sigma_{t,k} (as opposed with the two-level aggregation procedure which first computes an aggregated ranking for each task). Formally, for a system n, one compute $sum_n = \sum_(t,k) \sigma_{t,k}(n)$ where the sum is over each pair (t,k), and then rank the computed $sum_1, …, sum_N$.
> * In our implementation, the best system is ranked N (instead of 1) because when performing the usual _argsort_ method (in numpy), the highest item has the highest rank. Of course, we took that into account and drew our conclusions accordingly. In order to avoid confusion, we will update the implementation and reverse the _argsort_ to stick to the paper’s notations.
> Thus everything is reversed and for Borda Counts higher is better.
> * We will add the missing reference.
>
> _**We hope our answers address all concerns of reviewer qERs, in which case we hope they would be keen to consider raising their score.**_

---

### Official Review · Reviewer_BuYF · 2022-07-12

**Rating:** 4
**Confidence:** 3
**Soundness:** 2 fair
**Presentation:** 2 fair
**Contribution:** 2 fair

**Summary:**

This paper proposes a ranking method for comparing NLP systems that does not simply take the average of target scores. The authors propose to use a method called Kemeny consensus in which Kendall distance or correlation is used for choosing the best permutation (as a ranking candidate). The authors apply their method to simulated data, showing their ranking is robust to manipulation and scalable. In the experiments using real-world task sets such as GLUE and a set of NLG tasks, the authors show that their proposed ranking and mean-aggregation ranking correlate to some extent (but have some difference) and that their ranking is robust to adding and removing tasks.

**Questions:**

There are some grammatical and stylistic errors in the paper.

Minor:
- what is EXT in Table 2?

**Limitations:**

The authors only use NLG tasks for their instance-level analysis, but other types of tasks, such as machine translation and question answering, should be considered.

**Strengths And Weaknesses:**

This paper demonstrates that there might be a way of ranking systems in benchmarking that is different from simply taking the average of instance or task performance and comparing those scores. Such a ranking method may yield different outcomes in comparing systems, so it is highlighted that we need to be aware of the possibility of a different interpretation when doing the mean-aggregation comparison.

However, I do not understand the assumptions the authors have in this study. I have the following concerns:

- I think the paper fails to emphasize the necessity of ranking systems by aggregating benchmarking scores in some way. In other words, I do not think there should be the best ranking when benchmarking because the best model has to be determined depending on a pragmatic purpose.
- I am not sure if the proposed ranking method captures the intrinsic difficulty of tasks, which seems one of the main goals in this paper. When we interpret benchmarking results, the task difficulty needs to somehow be associated with human evaluation or external measures. However, in this study, the proposed ranking method only considers the relative difficulty regarding the evaluated systems.
- I do not necessarily think that ranking has to be robust to adding or removing tasks. Different systems may have different abilities that do not allow any sequential ranking of systems. Therefore, adding and removing tasks may change an ideal ranking.

I think these concerns weaken the substance of their experiments and contributions.

---

> ### Author Response · Authors · 2022-08-01
> **Reply to reviewer BuYF**
>
> First, let us thank reviewer BuYF for their thorough reading of the manuscript. In the following, we provide very concrete ways to alleviate their concerns that we will incorporate to an eventual camera-ready version of the manuscript. These points are also detailed in our response to reviewer 1ozF.
>
> **Final ranking must depend on the use case.** We totally agree with BuYF. Our setting was fixed so that reproducible experiments can be conducted. For specific use cases, a way to incorporate user preferences (such as “I only care about task A, D and H and task A is more important than the others”) is to consider a weighted version of the Borda count. It consists in choosing (this is to the user’s discretion) a weight $w_t$ for task t and then to compute the weighted Borda count $\sum_t w_t \eta_n^t$. That way, practitioners will have access to a satisfying, robust way to choose the best system for a given specific use case.
>
>
> **Taking intrinsic task difficulty into account.** The previously introduced weighted Borda count can help take intrinsic difficulty into account by fine-tuning the weights accordingly. Moreover, we propose to add another step in our methodology to allow for ties between systems. For a given task, we can compute the variance of the scores which somehow indicates its intrinsic difficulty. Then, if the scores difference between two systems on this task is below the computed variance, we consider these systems as equally performant and introduce a tie in the ranking.
>
> **Robustness to task addition is not relevant.** In our experimental setting, we implicitly assumed that each task was relevant. This allows us to study the robustness of the way systems are aggregated and to see that our ranking-based method is more robust. Of course, in real-life scenarios, only some sets of subtasks are relevant for specific use cases, but the same experiments could be conducted on this subset of tasks. We will add a remark on this in the manuscript.
>
> Two sum up, in order to take into account the concerns of reviewer BuYF, we will incorporate very concrete procedures in our methodology: (i) weighted Borda count and (ii) ties. This will allow us to take into account real-world scenarios and intrinsic difficulty of tasks.
>
> Again, we would like to thank reviewer BuYF for their relevant remarks that lead us to consider the aforementioned improvements of our methodology, which we think would really help NLP practitioners (for instance, our evaluation framework could consequently improve evaluation protocols of few shot classification, see for instance Logan et. al in ACL2022).
>
> _**We hope this properly resolves the concerns of reviewer BuYF and if so, that they will be keen to raise their score accordingly.**_
>
> **Reference**
> Logan IV, R. L., Balažević, I., Wallace, E., Petroni, F., Singh, S., & Riedel, S. (2021). Cutting down on prompts and parameters: Simple few-shot learning with language models. https://arxiv.org/pdf/2106.13353.pdf

---

### Official Review · Reviewer_1ozF · 2022-07-13

**Rating:** 6
**Confidence:** 4
**Soundness:** 3 good
**Presentation:** 3 good
**Contribution:** 3 good

**Summary:**

The paper proposes to address system benchmarking across several tasks as a ranking optimization problem and proposes a solution essentially based on aggregating preference rankings using Borda count. This is validated on synthetic data and using a large amount of real benchmarking on multiple NLP tasks, illustrating the benefits of the proposed method over simple averaging.


**Questions:**


1. Why is notation changing in Sec. 3.2., e.g. L tasks instead of T? Is this trying to convey something?
2. Please clarify the two argsorts on l.175?
3. Can you clarify how Borda count applies to KxT ranks over T tasks in one-level aggregation (l.188)?
4. Could you address the issue of ties mentioned in the previous section? How do you handle those?

Misc typos:

l.79: way, one -> way one

l.121: impracticable -> impractical

l.159: to -> two

l.162: Mallows statistical -> statistic?

l.172: it -- if in-line, verb agreement on l. 173-174 should be 3rd person, otherwise remove 'it'

l.205: where φ ∈ [0, 1] is a dispersion parameter [redundant with l.203]

Fig. 4 is oddly positioned on p.7 when it is referenced on p.9. Typo in caption: "Raning"

l.295: I think /sigma^{mean} should be /sigma^{mean,t}

l.326-334: Shouldn't Fig. 5 (b-e) and (g-j) be referred to somewhere here?

l.329: ?? [ref missing]

**Limitations:**

I don't see any potential negative societal impact of this work, beyond the potential negative societal impact the entire field of NLP may have. In fact, by addressing the sensitive issue of properly assessing progress and improvements in NLP system, this work may have positive impact and help discard spurious improvements that may actually harm society.

**Strengths And Weaknesses:**

*Strengths:*

The paper takes a different view on multiple benchmark aggregation, which is welcome especially in NLP, given that evaluation in the field is usually poorly performed and simplistic. The motivation is clear, the proposed solution (Borda count) is simple and effective. Its benefits are well argued and tested in an experimental section that uses results from a large amount of different tasks and systems.

*Weaknesses:*

Readability: Some figures (Figs. 2,3,4) are hard to read properly due to small font, even with on-screen magnification (Tab. 3 also). In addition, Figs. 3-5 are not self-contained, and not properly explained in the text either. As a consequence, the information the reader gets from these plots is sadly limited.

Also, The flip side of the strength above is that there is an awful lot of experimental results packed into the last two pages or so. So much in fact that it is difficult to get a clear picture of what is going on without referring to the supplementary material.

Limitation: One issue that is not addressed at all as far as I can tell is the handling of ties, an issue on which various variants of Borda counts differ. This is particularly relevant in this context for at least two reasons: First, proper benchmarking should consider significance (statistical or practical) of differences, and systems that are not significantly different would typically be considered as "tied". Second, many NLP tasks (e.g. categorization) produce discrete outputs and discrete scores, which would therefore yield many similar instance-level scores.

---

> ### Author Response · Authors · 2022-08-01
> **Reply to reviewer 1ozF**
>
> We would like to thank reviewer 1ozF for her/his careful reading of the manuscript and its insightful remarks that will allow us to improve the quality of our work. Here are our answers to the identified weaknesses.
>
> *  **Readability.** Due to space limitation, we had to choose small sizes for our Figures. We agree it degradates their readability and will make sure they will be larger for the eventual camera-ready version of the paper (the additional page will allow us to resize all Figures properly). We will also enrich the captions of Figs 3-5 to make them self-contained.
>
> *  **Experiments understanding.** The principal conclusions of our experiments are underlined in each “Takeaways” paragraph. We will put them more prominently, for instance by defining new subsections for each sub-experiment.
>
> *  **Handling ties.** We agree with 1ozF and have actually a way to handle ties by incorporating two additional considerations in our methodology:
>     *  First, for a given task which outputs some scores $s_1, …, s_T$, we will compute the variance of these scores and, if the difference between two systems scores is below the variance, we will consider them as equally performant for the task. These systems will therefore have the same rank in our framework and every Borda computation will be done in the same way, the only difference being that the final output is a ranking that allows for ties.
>      * Second, since the variability of a task may reflect its intrinsic difficulty, we propose to add parameters, which users can choose and that take into account this intrinsic difficulty. Formally, we associate a weight $w_t$ for each task t, and compute a weighted-version of the Borda count $\sum_t w_t \eta_n^t$ where $\eta_n^t$ is the rank of system n on task t.
>
> _We hope these two concrete, actionable solutions answer the concerns of 1ozF._
>
>
> Reviewer 1ozF also raised specific questions:
> 1. The change of notation “L” into “T” is a typo we did not correct. All “L” shall be replaced by “T” which is the number of tasks (hence the choice of T for the notation).
> 2. The definition of $\eta$ given line 174 is clear: it is the ranking of $sum_1, …, sum_T$ computed above. We shall remove the “argsort(argsort(...))” which is only the algorithmic way to compute $\eta$ but is confusing at this place in the paper.
> 3. Let us clarify the one-level aggregation. Each pair (t,k) of task t and utterance k leads to a ranking $\sigma_{t,k}$ of the N systems. In the one-level aggregation procedure, we directly compute the Borda aggregation of all of these $\sigma_{t,k}$ (as opposed with the two-level aggregation procedure which first computes an aggregated ranking for each task). Formally, for a system n, one compute $sum_n = \sum_{(t,k)} \sigma_{t,k}(n)$ where the sum is over each pair (t,k), and then rank the computed $sum_1, …, sum_N$.
> 4. As explained above, the handling of ties is straightforward with our procedure and we thank you for suggesting that as it will improve the generality of our work. Indeed, when several systems share the same rank, Borda count can be performed in the exact same way, eventually leading to final rankings that allow for ties.
>
>
> All mentioned typos will be resolved in the eventual camera-ready version of the paper.
>
> _**We hope our answers address all concerns of reviewer 1ozF, in which case we hope they would be keen to consider raising their score.**_

---

### Meta-Review · Area_Chair_jpov · 2022-08-25

**Recommendation:** Accept
**Confidence:** Certain

**Metareview:**

The paper proposes to address system benchmarking across several tasks as a ranking optimization problem and proposes a solution essentially based on aggregating preference rankings using Borda count. This is validated on synthetic data and using a large amount of real benchmarking on multiple NLP tasks, illustrating the benefits of the proposed method over simple averaging. Reviewers questions regarding how ties are resolved, taking task difficulty, and use-case were addressed convincingly by the authors. I hope they revise their manuscript so that it is easier to follow, the results section is packed with information, too many graphs to the point where finding information is difficult. I would strongly urge the authors to remove some analyses in the appendix. It would be good to also tell us how the various systems are being evaluated.

**Award:**

No

---

### Decision · Program_Chairs · 2022-09-14

Accept